# GMAI-VL & GMAI-VL-5.5M:
# A LARGE VISION-LANGUAGE MODEL AND
# A COMPREHENSIVE MULTIMODAL DATASET
# TOWARDS GENERAL MEDICAL AI

## ABSTRACT

Despite significant advancements in general artificial intelligence, such as GPT-4, their effectiveness in the medical domain (general medical AI, GMAI) remains constrained due to the absence of specialized medical knowledge. To address this challenge, we present GMAI-VL-5.5M, a comprehensive multimodal medical dataset created by converting hundreds of specialized medical datasets into meticulously constructed image-text pairs. This dataset features comprehensive task coverage, diverse modalities, and high-quality image-text data. Building upon this multimodal dataset, we propose GMAI-VL, a general medical vision-language model with a progressively three-stage training strategy. This approach significantly enhances the model's ability by integrating visual and textual information, thereby improving its ability to process multimodal data and support accurate diagnosis and clinical decision-making. Experimental evaluations demonstrate that GMAI-VL achieves state-of-the-art results across a wide range of multimodal medical tasks, such as visual question answering and medical image diagnosis. Our contributions include the development of the GMAI-VL-5.5M dataset, the introduction of the GMAI-VL model, and the establishment of new benchmarks in multiple medical domains.

## 1 INTRODUCTION

Large-scale Vision-Language Models (LVLMs) have rapidly evolved in recent years, effectively integrating visual perception with language understanding by leveraging large-scale multimodal data, which enables them to capture complex visual and textual patterns and drive significant advancements in image recognition, natural language processing, and multimodal tasks. With the advancement of multimodal integration technologies, the demand for high-precision processing of diverse data types in the medical field has become increasingly critical. The ability to effectively integrate and analyze various data modalities, such as medical images, clinical text, and structured clinical records, is pivotal for achieving accurate and comprehensive diagnostic and treatment decisions.

However, existing LVLMs, such as GPT-4 (Achiam et al., 2023), are limited in medical applications due to their lack of domain-specific knowledge, highlighting the need for specialized solutions that effectively integrate medical expertise. Addressing this challenge requires constructing a comprehensive medical vision-language dataset and developing domain-specific models. For the medical dataset, it should provide high-quality medical knowledge, including the following three aspects:

**Comprehensive Medical Task.** To enhance the model's applicability across various medical scenarios, the dataset should cover a wide range of medical contexts, such as disease types, symptoms, and treatments. Comprehensive task coverage can improve the model's generalization ability and increase its reliability in real-world applications. However, existing models often focus on specific domains (He et al., 2024; Li et al., 2024b; Xin Yan, 2023; Thawakar et al., 2024; Kapadnis et al., 2024), limiting their broader applicability. Expanding the dataset's scope will further enhance the model's utility in clinical practice.

**Rich Multimodal Representation.** A well-rounded medical multimodal dataset should encompass various modalities, including different medical imaging types (such as CT, MRI, and X-rays) and diverse forms of textual data (such as medical records and imaging reports). This would allow models to better integrate multi-source information and improve their analytical capabilities. However, existing methods tend to focus on a single type of medical imagery (Johnson et al., 2019; Wu et al., 2023b; Lu et al., 2024b), limiting the model's adaptability to diverse clinical scenarios. A more diverse multimodal dataset would provide a foundation for developing more comprehensive models, better suited to the complexity of real-world medical environments.

**High-Quality Image-Text Data.** High-quality training data is crucial for model performance. For medical applications, ideal image-text data should include a large collection of medical images with precise textual descriptions, enhancing the model's understanding of key medical concepts, including diagnosis, treatment, and clinical workflows, ultimately improving clinical outcomes. Although progress has been made by collecting data from sources like PubMed (Li et al., 2024a; Moor et al., 2023; Wu et al., 2023b), which has inconsistent data quality, imprecise alignment and lack of standardization, limiting its potential.

Based on above observations, we propose a methodology for developing a comprehensive multimodal medical datasets. The methodology begins by collecting large-scale, open-source medical imaging datasets and extracting key details such as modality, task type, labels, and bounding boxes. A vision-language model (*e.g.*, GPT-4o) is then used to transform these datasets, covering tasks like lesion detection, segmentation, and disease diagnosis, into high-quality image-text pairs for training LVLMs. To ensure data quality, extracted image information is incorporated into the prompt design, improving model performance across various clinical tasks. This results in a comprehensive multimodal dataset with 5.5M samples, named GMAI-VL-5.5M, which supports the development of general medical LVLMs. Fig. 1(a) illustrates the sources, departments, modalities, task types, and instruction formats of the constructed dataset.

With the constructed GMAI-VL-5.5M dataset, we develop a general medical vision-language model, GMAI-VL. To enhance its integration of visual and linguistic features and its instruction-following abilities, a three-stage training strategy is proposed in this paper. Specifically, we sequentially implement shallow and deep alignments in the first two stages, gradually building associations between visual (medical images) and language (medical texts) elements from basic features to high-level semantics. Next, we fine-tune the model with cross-modal instructions, improving its understanding of visual-language interactions and instruction-following in complex tasks. With this strategy, GMAI-VL shows strong performance in medical tasks like visual question answering and medical image diagnosis, providing a solid foundation for advancing multimodal models in the medical field.

Our contributions are as follows:

- We propose a methodology for constructing the GMAI-VL-5.5M, a comprehensive vision-language dataset with extensive coverage of medical tasks, diverse multimodal representations, and high-quality image-text pairs, forming a robust foundation for model training.

- With GMAI-VL-5.5M, we propose a versatile medical vision-language model, named GMAI-VL. Our proposed three-stage training strategy enhances its ability to integrate visual and language features, significantly improving the abilities of instruction-following and generalization across various medical tasks.

- GMAI-VL outperforms previous models in multimodal question-answering tasks, including PMC-VQA and VQA-RAD, setting new benchmarks on OmniMedVQA, GMAI-MMBench, and the health and medicine subset of MMMU. Specifically, GMAI-VL achieves an average score of 88.48% on OmniMedVQA, 62.43% on the GMAI-MMBench *test* set, and 51.3% on the health and medicine subset of MMMU.

## 2 RELATED WORK

**Large-scale Medical Vision-language Datasets** of high quality and multiple modalities are the basis of Large Vision-Language Models (LVLMs) in the medical domain. While natural language and vision datasets are easily accessible online, biomedical datasets often focus on text or images only and many of them are limited to specific tasks or modalities, thus with unsatisfactory general-

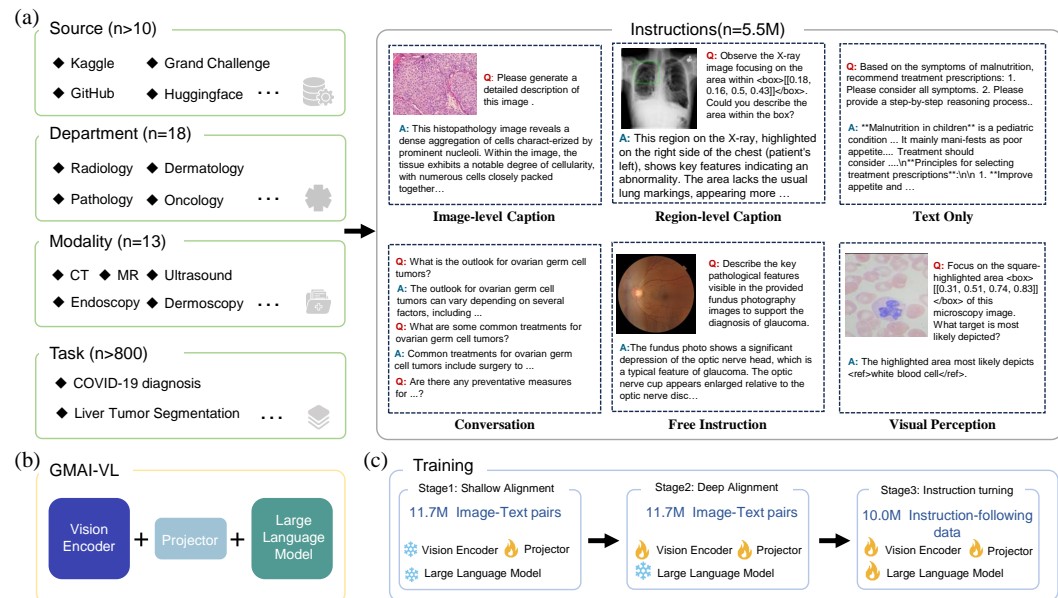

Figure 1: Overview of GMAI-VL and GMAI-VL-5.5M. (a) illustrates the sources, departments, modalities, task types, and instruction formats of the GMAI-VL-5.5M dataset. (b) Architecture of GMAI-VL, integrating a Vision Encoder, Projector, and Large Language Model. (c) Three-stage training process of GMAI-VL, including shallow alignment, deep alignment, and instruction tuning with corresponding data sizes and training components. The flame symbol 🔥 denotes the training part, while the snowflake symbol ❄ indicates frozen part.

ization ability. Notable datasets like MIMIC-CXR (Johnson et al., 2019) and CheXpert (Chambon et al., 2024) have advanced radiology models but are restricted to single image modality (X-ray), which hinders their use as general-purpose medical LVLMs. To address this, researchers have begun scraping public sources like PubMed and textbooks to construct large-scale vision-language datasets. Examples include datasets proposed in LLaVA-Med (Li et al., 2024a), Med-Flamingo (Moor et al., 2023), and PubMedVision (Chen et al., 2024b), with PubMedVision optimizing LLaVA-Med dataset for higher-quality medical data. In addition to scraping, open-source image datasets with annotations can also be converted into image-text pairs for model training. Specifically, image information like modalities and annotations are input into large language models, e.g., GPT series, to generate text paired with the corresponding image. Some popular examples include the datasets constructed in RadFM (Wu et al., 2023b), MedDr (He et al., 2024), MedTrinity-25M (Xie et al., 2024), ChiMed-VL (Liu et al., 2023b), BiomedGPT (Zhang et al., 2024), Med-Gemini (Saab et al., 2024), and Med-PaLM (Singhal et al., 2023).

These efforts usually suffer from either limited modalities, data sources, or task coverage. Thus, their dataset quality needs further improvement. To this end, we propose to construct a comprehensive medical vision-language dataset with extensive coverage of medical tasks, diverse multimodal representations, and high-quality image-text pairs, forming a robust foundation for model training.

**Medical Vision-Language Models** are usually based on general-purpose Large Vision-Language Models (LVLMs). Most of them adapt LVLMs to specific medical applications using specialized medical datasets. For instance, Med-Flamingo (Moor et al., 2023) enhances OpenFlamingo-9B using 0.8 million interleaved and 1.6 million paired medical image-text data, highlighting the critical need for multimodal data in medical image analysis and automated report generation tasks. RadFM (Wu et al., 2023b) improves PMC-LLaMA (Wu et al., 2023a) by leveraging 16 million radiology images with text descriptions from diverse sources. Similarly, Med-PaLM (Tu et al., 2024) adapts PaLM-E (Driess et al., 2023) to the medical domain with approximately one million medical data samples, achieving state-of-the-art performance in diagnostic support and medical knowledge Q&A. LLaVA-Med (Li et al., 2024a) utilizes a large-scale biomedical figure-caption dataset extracted from PubMed Central to enhance LLaVA (Touvron et al., 2023a;b) to better under-

Table 1: Comparison of various medical multimodal datasets, including details on the dataset size, modality type, language, data traceability, and sources of information.

| Datasets | Data Size | Modality | Language | Traceability | Data Source |
|---|---|---|---|---|---|
| PathVQA (He et al., 2020) | 32.7k | Pathology | EN | × | Textbooks |
| MIMIC-CXR (Johnson et al., 2019) | 227k | X-Ray | EN | ✓ | Hospital |
| quilt-1M (Ikezogwo et al., 2024) | 1M | Pathology | EN | × | YouTube & PubMed |
| MedDr VQA (He et al., 2024) | 197k | Multimodal | EN | ✓ | 13 medical datasets |
| PMC-OA (Lin et al., 2023) | 1.65M | Multimodal | EN | × | PubMed |
| PMC-VQA (Zhang et al., 2023) | 413k | Multimodal | EN | × | PubMed |
| LLaVA-Med VQA (Li et al., 2024a) | 56,702 | Multimodal | EN | × | PubMed |
| ChiMed-VL (Liu et al., 2023b) | 1.05M | Multimodal | CN | × | PubMed |
| PMC-CaseReport (Wu et al., 2023b) | 438k | Multimodal | EN | × | PubMed |
| PubMedVision (Chen et al., 2024b) | 1.29M | Multimodal | EN&CN | × | PubMed |
| **GMAI-VL-5.5M (ours)** | 5.5M | Multimodal | EN&CN | ✓ | 219 specialized medical imaging datasets |

stand biomedical images and facilitate open-ended conversational interactions. Med-Gemini (Saab et al., 2024) leverages long-format question-answering datasets to improve the multimodal and long-contextual capabilities of the baseline Gemini model, enabling superior performance in complex medical Q&A and multimodal reasoning tasks. Additionally, HuatuoGPT-Vision (Chen et al., 2024b) and MedDr (He et al., 2024) build medical multimodal datasets to adapt general-purpose LVLMs like LLaVA and InternVL to various medical modalities, including radiology, pathology, dermatology, and endoscopy.

Previous studies usually focus on constructing medical datasets to adapt general-purpose LVLMs but pay less attention to the adaptation strategies. However, naive training/adaptation strategies may not successfully adapt general-purpose LVLMs to the medical data, due to the large gap between the natural image-text pairs and the medical ones. Moreover, these strategies can hardly align the broad imaging modalities and various types of medical text (e.g., prescriptions, radiology reports, and electronic health records) to obtain generalizable features, thus limiting the models' performance. Our work thus proposes a novel three-stage training strategy to better integrate the visual and language features to enhance generalization ability.

## 3 GMAI-VL-5.5M: A COMPREHENSIVE MULTIMODAL DATASET

In the context of rapid advancements in medical vision-language models (VLMs), the construction of high-quality datasets is essential for developing general-purpose medical VLMs. Unlike previous methods that primarily rely on published literature to build medical vision-language datasets, our approach focuses on utilizing specialized medical datasets to develop a more robust and high-quality dataset. We introduce the GMAI-VL-5.5M, a comprehensive medical vision-language dataset that aggregates data from a wide range of sources, including both open-source and proprietary resources. The dataset encompasses 13 medical imaging modalities and covers 18 medical specialties, effectively addressing a broad spectrum of common medical imaging tasks. This dataset is designed to significantly enhance the model's capacity to understand and process complex medical information, thus contributing to advancements in precision medicine and intelligent diagnostics.

### 3.1 DATA CURATION

To construct a comprehensive multimodal medical dataset, we sourced 219 datasets from diverse platforms. Fig. 1(a) highlights key data sources, including Kaggle, Grand Challenge, and Huggingface, which enable extensive data collection. These datasets cover various imaging modalities, such as fundus, CT, MRI, and ultrasound (US), and span a range of medical tasks, including diagnosis, severity assessment, and organ recognition. Additionally, the datasets encompass multiple clinical departments, including pathology, dermatology, ophthalmology, otolaryngology, and oncology, further enhancing their diversity.

After data collection, we apply a preprocessing workflow to extract 2D medical images from the videos and 3D medical volumes. The preprocessed data are then standardized and organized into a

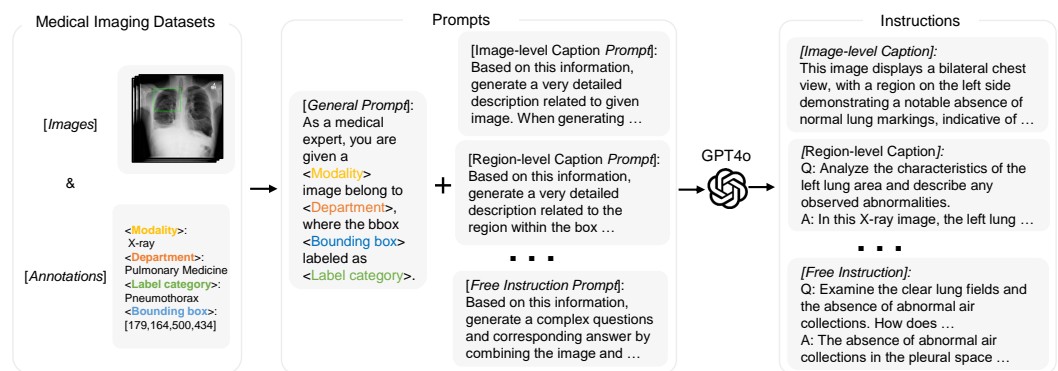

Figure 2: The proposed prompt-driven data generation methodology. Given a medical image, key annotation information is extracted into a structured format <image, modality, label, department, bbox [optional]> to generate a general prompt. Combined with general prompt, six specific prompts are desiged to produce six kinds of instruction-following data through GPT-4o.

structured format: <image, modality, label, department, bbox [optional]>. Subsequently, the data are categorized into two primary types: classification datasets and detection/segmentation datasets. Each category is further refined using specific prompts tailored for large model training. For data generation, large vision-language models (GPT-4o) are employed to produce detailed image descriptions and corresponding instruction-following data based on the designed prompts. For classification datasets, detailed descriptions of the entire image are generated, while for detection datasets, the focus is on specific regions enclosed by bounding boxes, providing comprehensive functional analyses of these areas. Notably, the segmentation dataset was transformed into a detection dataset using external bounding boxes, and data generation followed detection dataset protocols. Furthermore, to improve the model's multilingual capability, we translated a portion of English image-text data into Chinese. Incorporating multilingual data helps to enhance the generalization capabilities of domain-specific multimodal models. The resulted data is utilized for medical Visual Question Answering (VQA) tasks, forming the comprehensive VQA dataset, named GMAI-VL-5.5M. The detailed pipeline for generating prompt-driven data is illustrated in Fig. 2.

As depicted in Fig. 1(a), it contains six kinds of instrcuction-following formats, including image-level captions, region-level captions, free instructions, dialogue, visual perception and text-only tasks. The specific composition of GMAI-VL-5.5M can be found in Appendix (Table. 6). These formats enable VLMs to better understand and process complex visual and textual information in medical contexts. The GMAI-VL-5.5M dataset significantly enhances the model's cross-modal reasoning ability, enabling it to handle complex multimodal inputs in real clinical scenarios. The richness of the instruction formats allow the model to progress from basic question answering to advanced medical image analysis, ultimately providing strong support for clinical diagnosis and decision-making.

## 3.2 DATA PROPERTY

**Data Statistics.** These datasets encompass diverse medical imaging tasks and modalities, forming a solid foundation for developing and evaluating medical LVLMs. Fig. 3 illustrates the distribution of modalities, tasks, clinical departments, and specific medical challenges represented within the collected datasets. This visualization highlights the extensive diversity and coverage of our data collection efforts. After careful standardization and integration, these datasets form the core of our comprehensive medical image-text dataset, GMAI-VL-5.5M, which serves as a crucial resource for advancing precision medicine and intelligent diagnostic systems.

**Compared with other medical multimodal dataset.** The GMAI-VL-5.5M dataset, as highlighted in Table. 1, stands out due to its unmatched scale, encompassing over 5.5 million samples from more than 219 specialized medical imaging datasets. Unlike other datasets listed, GMAI-

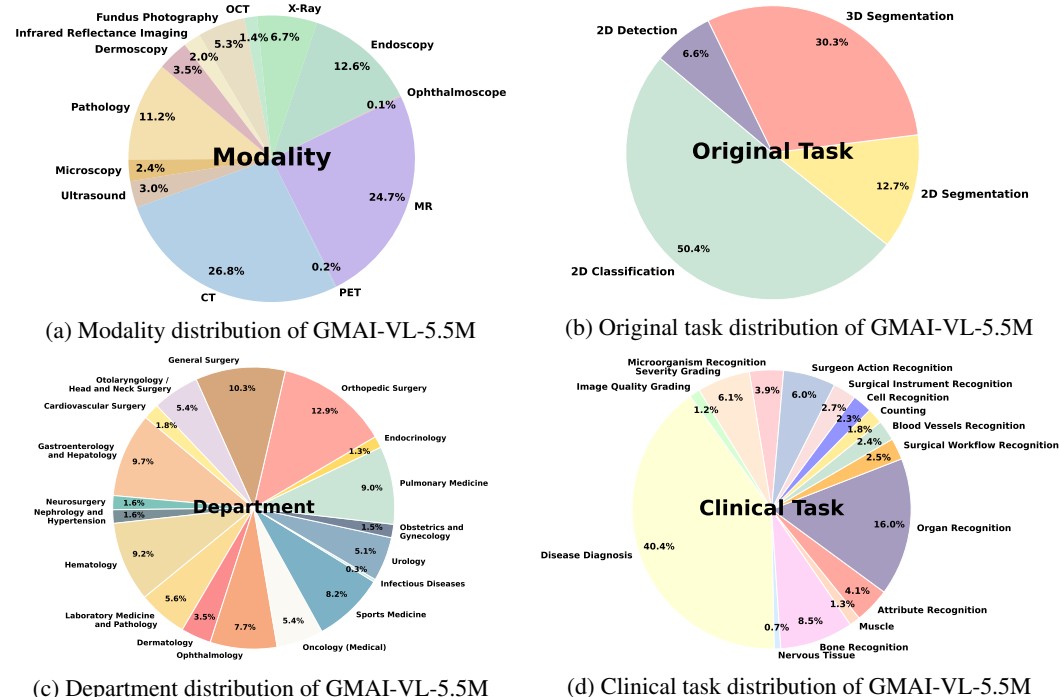

(a) Modality distribution of GMAI-VL-5.5M

(b) Original task distribution of GMAI-VL-5.5M

(c) Department distribution of GMAI-VL-5.5M

(d) Clinical task distribution of GMAI-VL-5.5M

Figure 3: Distribution of GMAI-VL-5.5M across tasks, modalities, departments, and clinical tasks. (a) Original Task Distribution: The dataset includes 2D Classification (50.4%), 3D Segmentation (30.3%), 2D Segmentation (12.7%), and 2D Detection (6.6%). (b) Modality Distribution: In addition to CT (26.8%) and MR (24.7%), X-ray (12.6%), Pathology (11.2%), and less common modalities like Dermoscopy (3.5%), Microscopy (2.4%), and PET (0.2%) are represented. (c) Department Distribution: While Orthopedic Surgery (12.9%) and General Surgery (10.3%) are the top contributors, departments like Endocrinology (1.3%), Infectious Diseases (0.8%), and Urology (0.7%) also provide data. (d) Clinical Task Distribution: Besides Disease Diagnosis (40.4%) and Organ Recognition (16.0%), tasks such as Muscle Recognition (3.3%), Nervous Tissue Recognition (1.5%), and Microorganism Recognition (1.2%) are included.

VL-5.5M supports a wider variety of modalities and languages, making it a truly global resource that caters to diverse clinical needs. Additionally, GMAI-VL-5.5M emphasizes traceability of its data, ensuring a high standard of clinical relevance and reliability. This comprehensive and diverse dataset is critical for pushing the boundaries of medical multimodal research, enabling more effective training of LVLMs that can generalize across multiple medical tasks and scenarios, thereby driving innovations in precision medicine and intelligent diagnostics.

# 4 GMAI-VL: A GENERAL MEDICAL VISION-LANGUAGE MODEL

## 4.1 ARCHITECTURE

The GMAI-VL model is a vision-language model built upon the LLaVA architecture (Liu et al., 2023a; Li et al., 2024a), incorporating three key components: a large language model (LLM), a vision encoder, and a projector (MLP), as illustrated in Fig. 1(b). These components are designed to work together seamlessly, enabling the model to deliver exceptional performance in medical applications.

We utilize InternLM2.5-7B (Team, 2023) as our language processing module, which offers outstanding reasoning capabilities. With a context window up to one million tokens, it can handle complex medical tasks and generate coherent, accurate responses. Its support for advanced instruction-following makes it particularly effective in addressing intricate medical queries, thereby enhancing the model's ability to understand and respond to a wide range of instructions.

For vision processing, GMAI-VL employs a CLIP-based vision encoder (Radford et al., 2021), which transforms visual inputs into high-dimensional feature representations. CLIP's strong performance in aligning image and text representations ensures that medical image features are accurately extracted and effectively integrated with linguistic information, significantly enhancing the model's ability to handle multimodal medical data.

The MLP, as a projector, serves as a bridge between the vision encoder and the LLM, optimizing high-dimensional outputs and further enhancing feature representation. The seamless integration of these components enables GMAI-VL to excel in processing and understanding complex medical multimodal data.

### 4.2 OPTIMIZATION STRATEGY

As illustrated in Fig. 1(c), the training process of the GMAI-VL model is divided into three stages: shadow alignment, deep alignment, instruction tuning, respectively. The detailed hyper-parameter settings can be found in Appendix (Table. 8). To enhance the training of GMAI-VL, we supplement our GMAI-VL-5.5M dataset with additional medical datasets. This supplemented data increases the diversity of training data, exposing the model to a wider range of medical scenarios and visual-language patterns, enhancing its generalization to complex clinical tasks and ensuring robustness in real-world applications. Fig. 4 (in Appendix) provides a complete distribution of the utilized datasets during the training stage. The detailed data proportions for each training stage are detailed in Table. 7 (in Appendix).

**Stage I: Shadow alignment.** In the shallow alignment phase, we utilize a large-scale medical image-text dataset comprising approximately 11.7 million image-text pairs, sourced from a combination of publicly accessible datasets and proprietary in-house data. To achieve shallow alignment, we freeze both the large language model and the vision encoder, optimizing only the projector. With this optimization stage, the model establishes an initial alignment between medical images and their corresponding textual descriptions. All input images are resized to $336 \times 336$ pixels, and the training objective is to minimize the cross-entropy loss of the text tokens.

**Stage II: Deep alignment.** Since most vision encoders in multimodal models are pre-trained on natural images, we address the domain differences between medical and natural images in the deep alignment stage. In this stage, we fine-tune both the vision-language projector and the vision encoder to achieve better alignment between the visual features of medical images and the feature space of the language model.

**Stage III: Instruction tuning.** At this stage, we fine-tune our GMAI-VL model (including the vision encoder, the language model, and the projector parts) by instruction tuning to enhance its instruction-following and dialogue capabilities. The multimodal instruction tuning data is primarily derived from the training data in previous stages, by filtering high-quality and more suitable data for fine-tuning. Additionally, we incorporate medical text dialogue data to ensure the model's versatility in handling various dialogue scenarios. Thus, our instruction tuning data comprises approximately 10 million samples.

## 5 EXPERIMENTS

To evaluate our model, we employed several established multimodal medical benchmarks, each targeting specific aspects of medical image understanding and question answering. Below is a brief overview of the benchmarks used in our experiments:

- **Traditional Medical VQA Benchmarks**: Traditional multimodal medical question-answering benchmarks, such as VQA-RAD (Lau et al., 2018), SLAKE (Liu et al., 2021), and PM-CVQA (Zhang et al., 2023), span various imaging modalities tasks, primarily assessing the model's ability to extract information from medical images and answer clinical questions. They evaluate the model's performance in understanding medical imaging and integrating multimodal information.

- **OmniMedVQA**: OmniMedVQA (Hu et al., 2024) provides a rich dataset of paired medical images and text, designed to evaluate the model's ability to recognize and understand fundamental medical imaging concepts, with a particular focus on cross-modal reasoning and information integration.

- **GMAI-MMBench**: GMAI-MMBench (Chen et al., 2024c) focuses on assessing the model's ability to identify fine-grained objects in complex clinical scenarios, challenging its capacity to handle long-context tasks and accurately recognize and reason over detailed medical features.

- **Health and Medicine Subset of the MMMU**: The health and medicine subset of the MMMU (Yue et al., 2024) benchmark spans a wide range of medical fields, derived from university exams, quizzes, and textbooks. It evaluates the model's reasoning ability in complex medical scenarios and the specialized knowledge in health and medicine.

Fuethermore, we present several examples of our model's performance across various tasks in Fig. 5 (Appendix).

## 5.1 EXPERIMENTS ON TRADITIONAL MEDICAL VQA BENCHMARKS

Table 2: Results on the Traditional Medical VQA Benchmarks. The best performance in each column is highlighted in red, and the second-best performance is highlighted in blue.

| Model | VQA-RAD | SLAKE | PMC-VQA | Avg. |
|---|---|---|---|---|
| Med-Flamingo (Moor et al., 2023) | 45.4 | 43.5 | 23.3 | 37.4 |
| RadFM (Wu et al., 2023b) | 50.6 | 34.6 | 25.9 | 37.0 |
| LLAVA-Med-7B (Li et al., 2024a) | 51.4 | 48.6 | 24.7 | 41.6 |
| Qwen-VL-Chat (Bai et al., 2023) | 47.0 | 56.0 | 36.6 | 46.5 |
| Yi-VL-34B (Young et al., 2024) | 53.0 | 58.9 | 39.5 | 50.5 |
| LLAVA-v1.6-7B (Liu et al., 2024) | 52.6 | 57.9 | 35.5 | 48.7 |
| LLAVA-v1.6-13B (Liu et al., 2024) | 55.8 | 58.9 | 36.6 | 50.8 |
| LLAVA-v1.6-34B (Liu et al., 2024) | 58.6 | 67.3 | 44.4 | 56.8 |
| HuatuoGPT-Vision-7B (Chen et al., 2024b) | 63.8 | 74.5 | 52.7 | **63.7** |
| **GMAI-VL(ours)** | 66.3 | 72.9 | 54.3 | 64.5 |

The performance of various VLMs on popular medical VQA benchmark datasets is summarized in Table. 2, including VQA-RAD(Lau et al., 2018), SLAKE(Liu et al., 2021), and PMC-VQA(Zhang et al., 2023). Our model, GMAI-VL, demonstrates a strong performance, achieving the highest score on the VQA-RAD(Lau et al., 2018) dataset with 66.3%, outperforming other models such as HuatuoGPT-Vision-7B. This result highlights GMAI-VL's superior capability in handling radiological image question-answering tasks. For the PMC-VQA(Zhang et al., 2023) dataset, GMAI-VL achieves 54.3%, and 72.9% on SLAKE(Liu et al., 2021), demonstrating its capability in handling medical VQA tasks across diverse modalities.

In conclusion, GMAI-VL demonstrates competitive performance across multiple benchmarks, showcasing its versatility in medical image understanding and question-answering.

## 5.2 EXPERIMENT ON OMNIMEDVQA

Table. 3 summarizes the performance of various large vision-language models (LVLMs), including our proposed GMAI-VL, across five question types: Modality Recognition, Anatomy Identification, Disease Diagnosis, Lesion Grading, and Other Biological Attributes. GMAI-VL demonstrates outstanding accuracy across multiple tasks, achieving 98.64% in Modality Recognition, 92.95% in Anatomy Identification, and 88.71% in Disease Diagnosis. It outperforms both open-source LVLMs and medical-specific models, underscoring its capability to accurately identify anatomical structures and diagnose diseases from visual data. In Lesion Grading, GMAI-VL attained the highest score of 87.21%, and it also delivers a strong performance of 82.95% in Other Biological Attributes, showcasing its versatility across diverse biological contexts. With an average accuracy of 88.48%, the highest among all evaluated models, GMAI-VL excels not only in general medical question-answer

Table 3: Comparison of performance between representative LVLMs and GMAI-VL on OmniMed-VQA across five different question type. The best performance in each column is highlighted in red, and the second-best performance is highlighted in blue.

| Model | Modality Recognition | Anatomy Identification | Disease Diagnosis | Lesion Grading | Other Biological Attributes | Overall |
|---|---|---|---|---|---|---|
| Random Guess | 25.00 | 25.84 | 28.41 | 25.40 | 37.49 | 28.28 |
| Open-Source LVLMs | | | | | | |
| MiniGPT-4 (Zhu et al., 2023) | 36.98 | 32.68 | 24.19 | 20.45 | 26.14 | 27.59 |
| LLaVA (Liu et al., 2023a) | 52.30 | 35.27 | 11.80 | 9.77 | 24.70 | 22.86 |
| LLaMA_Adapter_v2 (Gao et al., 2023) | 58.45 | 38.18 | 29.12 | 23.73 | 30.97 | 35.08 |
| InstructBLIP (Dai et al., 2024) | 72.35 | 39.90 | 32.01 | 43.80 | 47.91 | 41.14 |
| BLIP-2 (Li et al., 2023) | 57.48 | 49.83 | 46.21 | 30.52 | 73.52 | 50.77 |
| Qwen-VL-Chat (Bai et al., 2023) | 33.69 | 10.95 | 16.27 | 6.71 | 41.68 | 20.29 |
| mPLUG-Owl2 (Ye et al., 2023) | 78.01 | 48.52 | 39.68 | 20.56 | 59.36 | 48.44 |
| LLaVa-NeXT (Liu et al., 2024) | 68.23 | 46.74 | 41.21 | 18.43 | 39.57 | 45.57 |
| DeepSeek-VL (Lu et al., 2024a) | 74.01 | 51.94 | 45.46 | 21.06 | 29.04 | 48.76 |
| Yi-VL (Young et al., 2024) | 59.56 | 44.81 | 48.97 | 32.93 | 24.63 | 47.28 |
| InternVL2-40B (Chen et al., 2024d) | 96.76 | 64.25 | 76.28 | 76.50 | 76.27 | 78.70 |
| Medical Special Model | | | | | | |
| MedVInT-TE (Zhang et al., 2023) | 62.62 | 41.03 | 40.57 | 12.17 | 45.17 | 43.83 |
| LLaVA-Med (Li et al., 2024a) | 48.41 | 27.96 | 23.72 | 16.10 | 21.94 | 27.82 |
| Med-Flamingo (Moor et al., 2023) | 26.74 | 25.10 | 23.80 | 28.04 | 16.26 | 23.82 |
| RadFM (Wu et al., 2023b) | 27.45 | 21.65 | 23.75 | 16.94 | 20.05 | 23.48 |
| MedDr (He et al., 2024) | 91.37 | 51.62 | 65.56 | 73.18 | 74.52 | 68.27 |
| HuatuoGPT-Vision-34B (Chen et al., 2024b) | 95.06 | 75.67 | 66.51 | 72.83 | 74.92 | 73.23 |
| Our Model | | | | | | |
| **GMAI-VL(ours)** | 98.64 | 92.95 | 88.7 | 87.21 | 82.95 | 88.48 |

tasks but also in complex reasoning requiring domain-specific knowledge, surpassing models like HuatuoGPT-Vision-34B and InternVL2-40B.

These results verify our GMAI-VL is a leading model in multimodal medical image understanding, setting a new benchmark for medical VQA tasks. Its consistent top performance across question types highlights its potential for broader applications in medical question answering.

## 5.3 EXPERIMENTS ON MMMU HEALTH & MEDICINE TRACK

Table 4: Performance on the *val* set for the MMMU Health & Medicine track. This track is divided into five categories: **BMS** (*Basic Medical Science*), **CM** (*Clinical Medicine*), **DLM** (*Diagnostics and Laboratory Medicine*), **P** (*Pharmacy*), and **PH** (*Public Health*). The best performance in each column is highlighted in red, and the second-best performance is highlighted in blue.

| Model | BMS | CM | DLM | P | PH | MMMU Health & Medicine |
|---|---|---|---|---|---|---|
| Med-Flamingo (Moor et al., 2023) | 33.6 | 30.2 | 23.3 | 29.3 | 25.8 | 28.4 |
| RadFM (Wu et al., 2023b) | 31.6 | 28.6 | 26.7 | 26.2 | 26.8 | 27.9 |
| LLaVA-Med-7B (Li et al., 2024a) | 33.8 | 32.3 | 26.7 | 40.7 | 43.3 | 38.6 |
| Qwen-VL-Chat (Bai et al., 2023) | 32.7 | 20.6 | 19.3 | 29.6 | 33.3 | 31.7 |
| Yi-VL-34B (Young et al., 2024) | 48.1 | 55.6 | 36.7 | 35.4 | 31.3 | 48.2 |
| LLaVA-v1.6-7B (Liu et al., 2023a) | 46.4 | 43.4 | 30.0 | 29.6 | 26.7 | 33.1 |
| LLaVA-v1.6-13B (Liu et al., 2023a) | 53.6 | 46.7 | 33.3 | 22.2 | 40.0 | 39.3 |
| HuatouGPT-Vision-7B (Chen et al., 2024b) | 50.0 | 63.3 | 36.7 | 48.1 | 53.3 | 50.3 |
| **GMAI-VL(ours)** | 50.0 | 60.0 | 43.3 | 50.0 | 53.3 | 51.3 |

The MMMU benchmark, a widely recognized standard for evaluating multimodal models, was utilized to assess our proposed GMAI-VL model on the Health & Medicine track. The experimental results, presented in Table 4, show the model's performance across five key categories: Basic Medical Science (**BMS**), Clinical Medicine (**CM**), Diagnostics and Laboratory Medicine (**DLM**), Pharmacy (**P**), and Public Health (**PH**). GMAI-VL performs strongly across multiple categories, achieving top scores in **DLM** (43.3%), **P** (50.0%), and **PH** (53.3%), surpassing competitive models like LLaVA-v1.6 and HuatuoGPT-Vision-7B. These results highlight the model's proficiency in handling complex tasks requiring diagnostic reasoning, pharmaceutical knowledge, and public health expertise. In **BMS**, GMAI-VL scores 50.0%, achieve the state-of-the-art performance, demonstrating the model's the capacity of understanding medical knowledge. In **CM**, the model achieves

60.0%, remaining competitive with other leading models. These results underscore the model's ability in processing both clinical and foundational medical information effectively.

Overall, GMAI-VL achieves an average score of 51.3% across the Health & Medicine track, which is a top performance among other models, verifying its versatility in specialized medical domains.

## 5.4 EXPERIMENTS ON GMAI-MMBENCH

Table 5: Results on the *val* and *test* sets of GMAI-MMBench for clinical VQA tasks. The full names of the evaluated tasks can be found in Table.5 in literature (Chen et al., 2024c). The best model in each category is highlighted in red, while the second-best model is indicated in blue.

| Model Name | Overall (val) | Overall (test) | AR | BVR | B | CR | C | DD | IQG | MR | M | NT | OR-A | OR-HN | OR-P | OR-T | SG | SAR | SIR | SWR |
|---|---|---|---|---|---|---|---|---|---|---|---|---|---|---|---|---|---|---|---|---|
| Random Guess | | | | | | | | | | | | | | | | | | | | |
| Random | 25.70 | 25.94 | 38.20 | 22.73 | 22.92 | 22.72 | 24.06 | 26.66 | 27.13 | 27.00 | 20.00 | 24.75 | 21.37 | 22.93 | 22.33 | 21.18 | 32.43 | 24.23 | 21.39 | 23.71 |
| Medical Special Model | | | | | | | | | | | | | | | | | | | | |
| Med-Flamingo (Moor et al., 2023) | 12.74 | 11.64 | 6.67 | 10.14 | 9.23 | 11.27 | 6.62 | 13.43 | 12.15 | 6.38 | 8.00 | 18.18 | 9.26 | 18.27 | 11.00 | 11.53 | 12.16 | 5.19 | 8.47 | 11.43 |
| LLaVA-Med (Li et al., 2024a) | 20.54 | 19.60 | 24.51 | 17.83 | 17.08 | 19.86 | 15.04 | 19.81 | 20.24 | 21.51 | 13.20 | 15.15 | 20.42 | 23.73 | 17.67 | 19.65 | 21.70 | 19.81 | 14.11 | 20.86 |
| Qilin-Med-VL-Chat (Liu et al., 2023b) | 22.34 | 22.06 | 29.57 | 19.41 | 16.46 | 23.79 | 15.79 | 24.19 | 21.86 | 16.62 | 7.20 | 13.64 | 24.00 | 14.67 | 12.67 | 15.53 | 26.13 | 24.42 | 17.37 | 25.71 |
| RadFM (Wu et al., 2023b) | 22.95 | 22.93 | 27.16 | 20.63 | 13.23 | 19.14 | 20.45 | 24.51 | 23.48 | 22.85 | 15.60 | 16.16 | 14.32 | 24.93 | 17.33 | 21.53 | 29.73 | 17.12 | 19.59 | 31.14 |
| MedDr (He et al., 2024) | 41.95 | 43.69 | 41.20 | 50.70 | 37.85 | 29.87 | 28.27 | 52.53 | 36.03 | 31.45 | 29.60 | 47.47 | 33.37 | 51.33 | 32.67 | 44.47 | 35.14 | 25.19 | 25.58 | 32.29 |
| Open-Source LVLMs | | | | | | | | | | | | | | | | | | | | |
| Flamingo v2 (Awadalla et al., 2023) | 25.58 | 26.34 | 37.74 | 21.50 | 20.62 | 22.00 | 22.41 | 27.29 | 25.91 | 27.45 | 18.00 | 28.79 | 25.16 | 22.13 | 22.00 | 22.00 | 34.61 | 22.88 | 20.44 | 27.43 |
| VisualGLM-6B (Ding et al., 2021) | 29.58 | 30.45 | 40.16 | 33.92 | 24.92 | 25.22 | 24.21 | 32.99 | 29.96 | 29.53 | 21.20 | 37.88 | 30.32 | 24.80 | 13.33 | 29.88 | 33.11 | 19.62 | 19.16 | 37.43 |
| InstructBLIP-7B (Dai et al., 2024) | 31.80 | 30.95 | 42.12 | 26.92 | 24.92 | 28.09 | 21.65 | 34.58 | 31.58 | 29.23 | 22.40 | 30.30 | 28.95 | 27.47 | 23.00 | 24.82 | 32.88 | 19.81 | 21.64 | 26.57 |
| Qwen-VL (Bai et al., 2023) | 34.80 | 36.05 | 37.05 | 37.24 | 35.85 | 28.98 | 24.81 | 43.60 | 24.70 | 30.12 | 19.20 | 44.44 | 29.68 | 31.87 | 25.00 | 31.18 | 30.26 | 21.54 | 20.10 | 26.86 |
| Yi-VL-6B (Young et al., 2024) | 34.82 | 34.31 | 41.66 | 39.16 | 26.62 | 30.23 | 31.88 | 38.01 | 26.72 | 24.93 | 25.20 | 37.37 | 29.58 | 31.20 | 32.33 | 30.59 | 36.71 | 24.81 | 23.18 | 31.43 |
| LLaVA-NeXT-vicuna-7B (Liu et al., 2024) | 34.86 | 35.42 | 40.62 | 38.64 | 21.08 | 35.42 | 23.91 | 41.22 | 32.39 | 28.04 | 20.53 | 44.95 | 27.92 | 34.98 | 20.22 | 32.82 | 33.63 | 23.08 | 25.06 | 34.86 |
| CogVLM-Chat (Wang et al., 2023) | 35.23 | 36.08 | 40.97 | 30.77 | 27.69 | 32.74 | 19.40 | 41.10 | 36.84 | 34.72 | 24.00 | 40.91 | 36.74 | 37.33 | 26.00 | 33.65 | 36.56 | 20.19 | 23.95 | 26.57 |
| Monkey (Li et al., 2024d) | 35.48 | 36.39 | 38.32 | 35.31 | 35.54 | 34.53 | 23.16 | 43.40 | 31.98 | 30.12 | 19.20 | 33.33 | 30.00 | 32.53 | 25.33 | 31.65 | 34.46 | 20.00 | 20.27 | 30.29 |
| mPLUG-Owl2 (Ye et al., 2023) | 35.62 | 36.21 | 37.51 | 41.08 | 30.92 | 38.10 | 27.82 | 41.59 | 28.34 | 32.79 | 22.40 | 40.91 | 24.74 | 38.27 | 23.33 | 36.59 | 33.48 | 20.58 | 23.01 | 32.86 |
| ShareGPT4V-7B (Chen et al., 2023a) | 36.71 | 36.70 | 43.96 | 37.59 | 21.54 | 37.57 | 18.80 | 43.26 | 32.39 | 27.30 | 22.80 | 43.43 | 29.47 | 37.33 | 22.00 | 31.76 | 34.98 | 24.42 | 25.06 | 30.00 |
| InternVL-Chat-V1.1 (Chen et al., 2023b) | 38.16 | 39.41 | 42.46 | 43.88 | 35.23 | 45.08 | 23.31 | 45.96 | 38.87 | 29.23 | 29.60 | 40.40 | 31.68 | 41.87 | 26.67 | 38.82 | 32.13 | 19.42 | 25.58 | 30.29 |
| LLAVA-V1.5-7B (Liu et al., 2023a) | 38.23 | 37.96 | 45.45 | 34.27 | 30.92 | 41.32 | 21.65 | 44.68 | 34.01 | 27.74 | 23.60 | 43.43 | 28.00 | 42.13 | 29.00 | 35.06 | 33.41 | 22.12 | 23.61 | 29.14 |
| XComposer2 (Dong et al., 2024) | 38.68 | 39.20 | 41.89 | 37.59 | 33.69 | 40.79 | 22.26 | 45.87 | 36.44 | 32.94 | 27.20 | 58.59 | 26.11 | 36.40 | 43.67 | 37.29 | 32.06 | 23.46 | 27.80 | 32.86 |
| LLAVA-InternLM-7b (Contributors, 2023) | 38.71 | 39.11 | 36.36 | 36.54 | 32.62 | 38.10 | 30.68 | 46.53 | 34.82 | 28.19 | 25.20 | 48.99 | 28.11 | 40.53 | 33.33 | 36.00 | 34.08 | 26.73 | 24.12 | 29.71 |
| InternVL-Chat-V1.5 (Chen et al., 2024d) | 38.86 | 39.73 | 43.84 | 44.58 | 34.00 | 33.99 | 31.28 | 45.59 | 33.20 | 38.28 | 32.40 | 42.42 | 31.89 | 42.80 | 27.00 | 36.82 | 34.76 | 23.27 | 24.72 | 32.57 |
| InternVL-Chat-V1.2 (Chen et al., 2023b) | 39.52 | 40.01 | 41.66 | 44.06 | 27.38 | 38.46 | 34.29 | 46.99 | 33.60 | 34.42 | 21.20 | 47.98 | 30.63 | 42.80 | 27.67 | 35.88 | 35.59 | 23.85 | 24.98 | 28.00 |
| LLAVA-InternLM2-7b (Contributors, 2023) | 40.07 | 40.45 | 39.82 | 37.94 | 30.62 | 35.24 | 29.77 | 48.97 | 34.01 | 25.96 | 20.80 | 53.03 | 30.95 | 42.67 | 32.00 | 39.88 | 32.43 | 21.73 | 24.38 | 38.00 |
| DeepSeek-VL-1.3B (Lu et al., 2024a) | 40.25 | 40.77 | 38.55 | 35.14 | 38.92 | 40.07 | 27.97 | 48.12 | 35.63 | 31.75 | 22.80 | 46.97 | 40.74 | 44.93 | 31.00 | 40.47 | 33.33 | 22.31 | 21.39 | 31.71 |
| DeepSeek-VL-7B (Lu et al., 2024a) | 41.73 | 43.43 | 38.43 | 47.03 | 42.31 | 37.03 | 26.47 | 51.11 | 33.20 | 31.16 | 26.00 | 44.95 | 36.00 | 58.13 | 36.33 | 47.29 | 34.91 | 18.08 | 25.49 | 39.43 |
| MiniCPM-V2 (Xu et al., 2024) | 41.79 | 42.54 | 40.74 | 43.01 | 36.46 | 37.57 | 27.82 | 51.08 | 28.74 | 29.08 | 26.80 | 47.47 | 37.05 | 46.40 | 25.33 | 46.59 | 35.89 | 22.31 | 23.44 | 31.71 |
| Proprietary LVLMs | | | | | | | | | | | | | | | | | | | | |
| Claude3-Opus (Anthropic, 2024) | 32.37 | 32.44 | 1.61 | 39.51 | 34.31 | 31.66 | 12.63 | 39.26 | 28.74 | 30.86 | 22.40 | 37.37 | 25.79 | 41.07 | 29.33 | 33.18 | 31.31 | 21.35 | 23.87 | 4.00 |
| Qwen-VL-Max (Bai et al., 2023) | 41.34 | 42.16 | 32.68 | 44.58 | 31.38 | 40.79 | 10.68 | 50.53 | 32.79 | 44.36 | 29.20 | 51.52 | 41.37 | 58.00 | 30.67 | 41.65 | 26.95 | 25.00 | 24.64 | 39.14 |
| GPT-4V (Achiam et al., 2023) | 42.50 | 44.08 | 29.92 | 48.95 | 44.00 | 37.39 | 12.93 | 52.88 | 32.79 | 44.21 | 32.80 | 63.64 | 39.89 | 54.13 | 37.00 | 50.59 | 27.55 | 23.08 | 25.75 | 37.43 |
| Gemini 1.0 (Team et al., 2023) | 44.38 | 44.93 | 42.12 | 45.10 | 46.46 | 37.57 | 20.45 | 53.29 | 35.22 | 36.94 | 25.20 | 51.01 | 34.74 | 59.60 | 34.00 | 50.00 | 36.64 | 23.65 | 23.87 | 35.43 |
| Gemini 1.5 (Reid et al., 2024) | 47.42 | 48.36 | 43.50 | 56.12 | 51.23 | 47.58 | 2.26 | 55.33 | 38.87 | 48.07 | 30.00 | 76.26 | 51.05 | 75.87 | 46.33 | 62.24 | 20.57 | 27.69 | 30.54 | 40.57 |
| GPT-4o (Achiam et al., 2023) | 53.53 | 53.96 | 38.32 | 61.01 | 57.08 | 49.02 | 46.62 | 61.45 | 46.56 | 56.38 | 34.00 | 75.25 | 53.79 | 69.47 | 48.67 | 65.88 | 33.93 | 22.88 | 29.51 | 39.43 |
| Our Model | | | | | | | | | | | | | | | | | | | | |
| **GMAI-VL(ours)** | 61.74 | 62.43 | 75.26 | 59.66 | 67.24 | 56.86 | 54.29 | 67.14 | 42.80 | 79.97 | 41.60 | 75.00 | 60.45 | 75.48 | 53.33 | 58.12 | 42.09 | 72.31 | 37.40 | 59.14 |

The GMAI-MMBench benchmark is a comprehensive medical multimodal benchmark designed to evaluate models on a range of clinical visual question-answering (VQA) tasks. Table 5 presents the results of various LVLMs, including open-source LVLMs and commercial models, evaluated on the *val* and *test* sets across multiple clinical tasks. GMAI-VL outperforms other models, achieving the highest scores on both the *val* set with 59.23% and the *test* set with 59.89%, surpassing the leading commercial models such as GPT-4V and Gemini 1.5. Notably, GMAI-VL excels in specific tasks such as abnormality recognition (73.78%), biological variation recognition (63.06%), and clinical disease diagnosis (66.67%). These results demonstrates the model's strong ability in understanding and interpreting complex clinical images. In comparison to other models, GMAI-VL consistently achieves either the best or second-best performance across most tasks. For instance, it ranks first in 16 out of 20 categories, including key tasks such as AR (Attribute Recognition) and DD (Disease Diagnosis), where it achieved scores of 75.26% and 67.14%, respectively, suggesting GMAI-VL's strength in understanding medical scenarios.

Overall, GMAI-VL establishes a new benchmark in various clinical VQA tasks, demonstrating its potential as a reliable and versatile tool in medical multimodal applications.

## 6 CONCLUSION

In this study, we introduce GMAI-VL, a large vision-language model, along with GMAI-VL-5.5M, a comprehensive multimodal medical dataset aimed at advancing general medical AI (GMAI). GMAI-VL-5.5M, which converts hundreds of medical image analysis datasets into high-quality image-text pairs, enables GMAI-VL to effectively address a wide range of clinical tasks. Experimental results show that GMAI-VL-5.5M significantly enhances GMAI-VL's performance on diverse clinical tasks, achieving state-of-the-art results across several key benchmarks.

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

# Appendix

CONTENTS

# A APPENDIX

## A.1 DETAILS FOR GMAI-VL-5.5M

In Table. 6 , we provide sub-datasets information of the multimodal dataset GMAI-VL-5.5M we have constructed. Based on the different data forms introduced in the paper, we have categorized the data into five distinct sub-datasets. These include GMAI-MM-Caption-1.7M, GMAI-MM-Instruct-0.9M, GMAI-MM-Percept-1.3M, GMAI-Text-Single-1M, and GMAI-Text-0.6M. Each sub-dataset corresponds to specific components: image caption data, free instruction data, visual perception data, text-only, and conversation.

Table 6: Sub-Dataset Details for GMAI-VL-5.5M

| Dataset | Sub-Dataset Name | Description | Size |
|---|---|---|---|
| GMAI-VL-5.5M | GMAI-MM-Caption-1.7M | A curated set of detailed medical image captions. | 1.7M |
| | GMAI-MM-Instruct-0.9M | A diverse set of instructions for medical image analysis. | 0.9M |
| | GMAI-MM-Percept-1.3M | A dataset of labels for medical image classification and segmentation. | 1.3M |
| | GMAI-Text-Single-1M | A set of single-round medical dialogues on patient queries | 1.0M |
| | GMAI-Text-Multi-0.6M | A dataset of multi-turn medical conversations on various topics. | 0.6M |

## A.2 TRAINING DATA DETAILS

In this section, we provide a comprehensive overview of all datasets utilized for training the GMAI-VL model. The details include the dataset names, their corresponding categories, the amount of data used for training, and the proportion of training data allocated to each dataset during the three phases of model training.

Table 7 summarizes the datasets employed, highlighting their respective categories and sizes. It is important to note that for certain datasets, we performed data cleaning and bilingual translation. As a result, the dataset sizes reported here may differ from the official numbers.

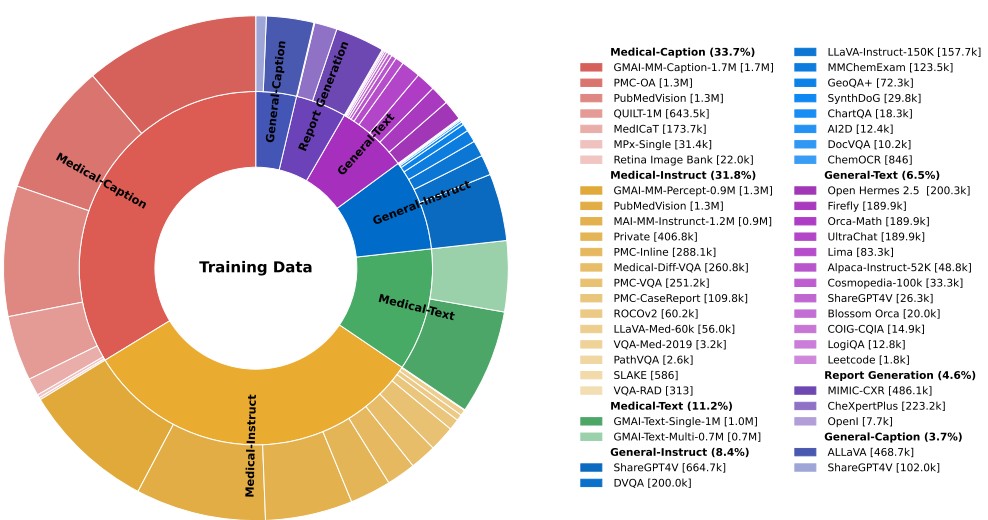

Figure 4: Distribution of Our Training Data.

Table 7: List of datasets used in our model. We employ a large collection of image-text data and instruction data for training stage.

| Dataset Category | Dataset Name | Size | ratio in stage 1&2 | ratio in stage 3 |
|---|---|---|---|---|
| General Captioning | ALLaVA(Chen et al., 2024a) | 468k | 100.0% | 50.0% |
| | ShareGPT4V(Chen et al., 2023a) | 102k | | |
| Medical Captioning | GMAI-MM-Caption-1.7M | 1.7M | 100.0% | 100.0% |
| | PubMedVision(Chen et al., 2024b) | 1.3M | | |
| | MedICaT(Subramanian et al., 2020) | 173k | 100.0% | 5.0% |
| | MPx-Single(Wu et al., 2023b) | 31k | | |
| | PMC-OA(Lin et al., 2023) | 1.3M | | |
| | QUILT-1M(Ikezogwo et al., 2024) | 643k | | |
| | Retina Image Bank(ASRS, 2024) | 22k | | |
| Report Generation | CheXpertPlus(Chambon et al., 2024) | 223k | 100.0% | 30.0% |
| | MIMIC-CXR(Johnson et al., 2019) | 486k | | |
| | OpenI(Demner-Fushman et al., 2016) | 7k | | |
| General Instruction | GeoQA+(Cao & Xiao, 2022) | 72k | 100.0% | 75.0% |
| | AI2D(Kembhavi et al., 2016) | 12k | | |
| | SynthDoG(Kim et al., 2022) | 29k | | |
| | ChartQA(Masry et al., 2022) | 18k | | |
| | MMChemExam(Li et al., 2024c) | 219k | | |
| | LLaVA-Instruct-150K(Liu et al., 2023a) | 157k | | |
| | DVQA(Kafle et al., 2018) | 200k | | |
| | DocVQA(Mathew et al., 2021) | 10k | | |
| Medical Instruction | GMAI-MM-Percept-1.3M | 1.3M | 100.0% | 100.0% |
| | GMAI-MM-Instruct-0.9M | 0.9M | | |
| | PubMedVision(Chen et al., 2024b) | 1.28M | | |
| | LLaVA-Med-60k(Li et al., 2024a) | 56k | | |
| | PMC-Inline(Wu et al., 2023b) | 288k | 100.0% | 10.0% |
| | VQA-Med-2019(Ben Abacha et al., 2019) | 3.2k | | |
| | Medical-Diff-VQA(Hu et al., 2023) | 260k | | |
| | PathVQA(He et al., 2020) | 2.6k | | |
| | PMC-CaseReport(Wu et al., 2023b) | 109k | | |
| | PMC-VQA(Zhang et al., 2023) | 251k | | |
| | ROCOV2(Rückert et al., 2024) | 60k | | |
| | SLAKE(Liu et al., 2021) | 0.6k | | |
| | VQA-RAD(Lau et al., 2018) | 0.3k | | |
| General Text | blossom_orca(Azure99, 2024) | 20k | 0.0% | 100.0% |
| | COIG-CQIA(Bai et al., 2024) | 14.8k | | |
| | Cosmopedia-100k(Ben Allal et al., 2024) | 33k | | |
| | ShareGPT4V(Chen et al., 2023a) | 26k | | |
| | Orca-Math(Mitra et al., 2024) | 379k | | |
| | Leetcode(Bernard, 2023) | 1.7k | | |
| | LogiQA(Liu et al., 2020) | 12.7k | | |
| | Lima(GAIR, 2023) | 83k | | |
| | Open Hermes 2.5(Teknium, 2023) | 200k | | |
| | Firefly(Yang, 2023) | 189k | | |
| | UltraChat(Ding et al., 2023) | 189k | | |
| | Alpaca-Instruct-52K(Taori et al., 2023) | 49k | | |
| Medical Text | GMAI-Text-Single-1M | 1.0M | 0.0% | 100.0% |
| | GMAI-Text-Multi-0.6M | 649k | | |
| **Overall** | - | **15.7M** | - | - |

Table 8: Training settings of GMAI-VL's stage I, stage II, and stage III.

| Settings | Stage I | Stage II | Stage III |
|---|---|---|---|
| freeze LLM | True | True | False |
| freeze MLP | False | False | False |
| freeze Vision Encoder | True | False | False |
| packing type | soft packing | soft packing | soft packing |
| learning rate | 1e-3 | 1e-4 | 1e-5 |
| learning rate schedule | cosine decay | cosine decay | cosine decay |
| optimizer | AdamW | AdamW | AdamW |
| optimizer hyper-parameters | $\beta_1 = 0.9, \beta_2 = 0.999$ | $\beta_1 = 0.9, \beta_2 = 0.999$ | $\beta_1 = 0.9, \beta_2 = 0.999$ |
| input size | 336x336 | 336x336 | 336x336 |
| total batch size | 32x8x2 | 32x4x4 | 32x4x4 |
| drop rate | 0.0 | 0.0 | 0.0 |
| numerical precision | DeepSpeed bf16 | DeepSpeed bf16 | DeepSpeed bf16 |
| GPUs for training | 32xA100 (80G) | 32xA100 (80G) | 32xA100 (80G) |

### A.3 MODEL TRAINING SETTINGS

The table. 8 presents the training settings for GMAI-VL across three stages, detailing key hyperparameters.

- Stage I (Shallow alignment). The large language model (LLM) is frozen, while the MLP is trainable, and the vision encoder is frozen. The learning rate is set to $1e^{-3}$ with a cosine decay schedule, using AdamW as the optimizer. Input size is $336 \times 336$, and the total batch size is $32 \times 8 \times 2$, with no dropout.

- Stage II (Deep alignment). Both the LLM and MLP remain frozen, but the vision encoder is unfrozen and trainable. The learning rate is lowered to $1e^{-4}$, and other settings remain consistent with Stage 1.

- Stage III (Instruction tuning). None of the components are frozen, allowing the entire model to be fine-tuned. The learning rate is further reduced to $1e^{-5}$, while other parameters, including optimizer and batch size, remain unchanged across stages.

Each stage utilizes DeepSpeed for mixed-precision training (bf16) on 32 A100 (80GB) GPUs.

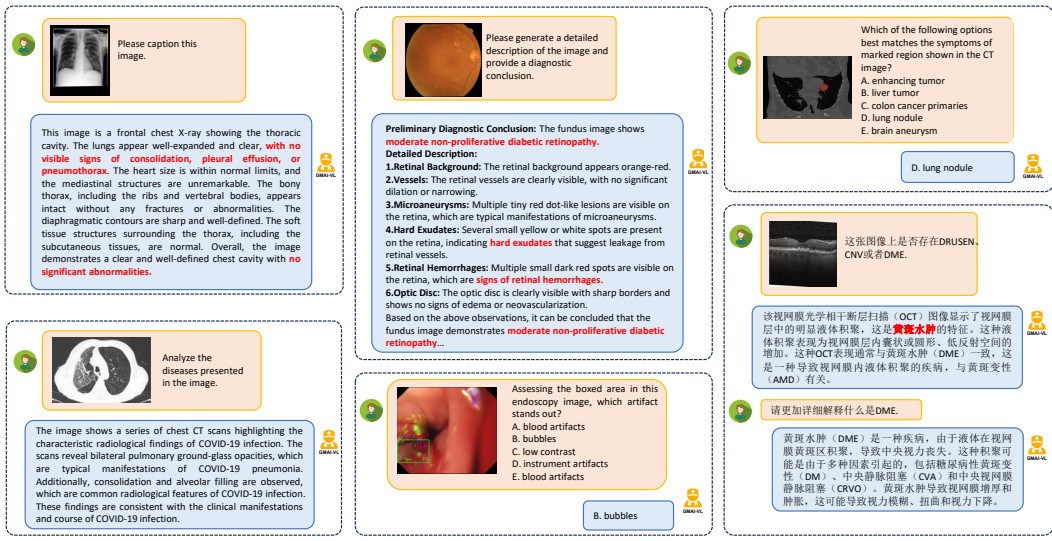

Figure 5: Examples of our experimental results.

## A.4 RESULTS

Fig. 5 illustrates several examples of our model on various tasks including image description, disease diagnosis, free question answering, etc., demonstrating that our multimodal model has strong general capabilities in the medical field. In addition, our model supports bilingual capabilities in Chinese and English.

