# OpenReview forum: "GMAI-VL & GMAI-VL-5.5M: A Large Vision-Language Model and  A Comprehensive Multimodal Dataset Towards General Medical AI"
_ICLR.cc/2025/Conference — ICLR 2025 Conference Withdrawn Submission_

### Official Review · Reviewer_WuaT · 2024-10-16

**Soundness:** 3
**Presentation:** 3
**Contribution:** 2
**Rating:** 3
**Confidence:** 3

**Summary:**

This article proposes a general medical visual language model named GMAI-VL and builds a large-scale multi-modal dataset containing 5.5 million samples (GMAI-VL-5.5M). The goal of the article is to make up for the shortcomings of current general artificial intelligence in the medical field and improve the model's performance in medical tasks through multi-modal data sets and a three-stage training strategy.

**Strengths:**

1. The data set covers 13 medical imaging modalities and 18 clinical fields, with diverse data sources (such as Kaggle, Huggingface) and broad coverage.
2. 219 professional medical data sets were converted and standardized into image-text pairs to ensure high quality and diversity, and some data support Chinese and English, improving the cross-language adaptability of the model.

**Weaknesses:**

1. Reasonableness of Data and Control of Noisy Information
 The GMAI-VL-5.5M dataset heavily relies on open-source data, which can vary in quality and may contain inconsistent labels or noise, potentially disrupting the model's training process.  Provide a detailed description of the data cleaning steps, including how missing labels, inconsistent annotations, and potential noise were handled. A flowchart or reference code for the data preprocessing process can be added to the paper for better clarity.  Combine manual and automated review methods to assess annotation consistency. The results of consistency evaluation (e.g., annotation agreement metrics) should be reported.

2. Lack of Sufficient Evidence for Model Superiority
 Although the paper introduces the GMAI-VL model, relying solely on experimental results to demonstrate its superiority is insufficient. Since GMAI-VL is trained on a proprietary large dataset, while other medical large models have not been fine-tuned on the same data, the current experiments may not fully validate the claimed performance advantages.  Conduct comprehensive comparisons with other mainstream medical large models on public datasets to ensure fair and representative performance comparisons.  Fine-tune other models on the same proprietary dataset used for GMAI-VL, followed by performance comparisons to ensure that the superiority of GMAI-VL is well-validated.

3. Insufficient Justification for the Dataset as a Benchmark
  Although the study aims to develop a medical model with strong general capabilities, further experiments are needed to justify the conclusion that the comprehensive training model outperforms models trained on specialized tasks, such as Mixture of Experts (MoE) models. Include additional comparison experiments to evaluate the performance of GMAI-VL against MoE models specialized in corresponding tasks.  Emphasize the performance of GMAI-VL in cross-domain tasks, demonstrating its advantage in handling diverse tasks and proving that it can outperform specialized models in multiple fields.

**Questions:**

N/A

---

### Official Review · Reviewer_35c4 · 2024-10-31

**Soundness:** 2
**Presentation:** 2
**Contribution:** 2
**Rating:** 3
**Confidence:** 5

**Summary:**

The paper introduces GMAI-VL-5.5M, a comprehensive multimodal medical dataset comprising image-text pairs. Additionally, it presents GMAI-VL, a robust general medical vision-language model designed to effectively process multimodal data for clinical decision-making. Experimental results across four datasets demonstrate the model’s effectiveness and potential impact.

**Strengths:**

- The paper is well-written and easy to follow, with visualizations that effectively illustrate the architectural concepts.
- It introduces an open-source dataset encompassing multiple modalities and tasks, though questions remain regarding its overall usefulness.

**Weaknesses:**

Overall, I am concerned about the novelty of this work. Although the authors introduce a large-scale, open-source dataset, it has several notable limitations:
- All labels, masks, and instructions were generated by GPT-4, potentially restricting performance in downstream tasks. Notably, this dataset was not used for validation, which is unexpected.
- The dataset lacks a normalization or standardization process, suggesting that distributions may vary significantly across sources. This variability could have a substantial impact on model training.
- No clinical validation was conducted to confirm the dataset’s practical utility.

Additional Observations:
- In Section 3.1, given that GPT-4 was used to curate the dataset, one would expect downstream performance to be constrained by GPT-4’s limitations. It would be helpful to see GPT-4’s performance on this dataset as a baseline for comparison.
- Section 3 lacks critical details regarding data preparation. It is unclear how images from different institutions were standardized—specifically, the types of scanners used, the consistency in imaging perspectives, any handling of motion blur, and how such issues were managed. Additionally, information on dataset distribution, potential variability, and any applied thresholds is missing. These details are essential to determine the dataset’s suitability for models operating on diverse distributions.
- Although the authors prepared and open-sourced the GMAI-VL-5.5M dataset, it is surprising that metrics from the proposed benchmark are not reported.
- In Line 347, it is mentioned: “We utilize a large-scale medical image-text dataset comprising approximately 11.7 million image-text pairs.” Clarification is needed on whether this dataset is part of GMAI-VL-5.5M and how these data were sourced.
- In Line 362, regarding instruction tuning, the authors note: “The multimodal instruction tuning data is primarily derived from the training data in previous stages by filtering high-quality and more suitable data for fine-tuning.” However, it is unclear how this filtering process was conducted and what inclusion or exclusion criteria were applied.
- Line 320 raises another question: Why was InternLM2.5-7B chosen as the model base instead of a similarly scaled alternative, such as LLaMA-3-8B, which also functions as a generalist? Additionally, has Bio-LLM been considered for the language model?
- In Table 2, the superiority of the proposed method over other baselines, such as HuatouGPT-Vision-7B, remains unconvincing.
- No ablation study has been conducted to evaluate the contribution of each module within the proposed network.

**Questions:**

See weaknesses.

**Details Of Ethics Concerns:**

In line 348, the authors state, “In the shallow alignment phase, we utilize a large-scale medical image-text dataset comprising approximately 11.7 million image-text pairs, sourced from a combination of publicly accessible datasets and proprietary in-house data.” However, the in-house data is not mentioned elsewhere in the work, raising concerns about ethical compliance and data transparency.

---

### Official Review · Reviewer_HMaR · 2024-11-03

**Soundness:** 2
**Presentation:** 3
**Contribution:** 3
**Rating:** 5
**Confidence:** 4

**Summary:**

This work curated a large-scale medical vision-language dataset by gathering public medical datasets and prompting GPT4o to generate descriptions. A LLaVA-style multimodal model and a new benchmark were further developed based on the curated dataset.

**Strengths:**

- Comprehensive medical vision-language dataset with 5.5M image-text pairs, covering various diseases and modalities
- Based on the dataset, GMAI-VL model is developed for vision QA and diagnosis
- Provide a new benchmark for medical VLMs

**Weaknesses:**

- GPT4o is employed to produce detailed image descriptions and corresponding instruction-following data based on the designed prompts. However, GPT-4's performance in identifying specific pathologies is inconsistent. https://link.springer.com/article/10.1007/s00330-024-11035-5

- Experiments: lack of ablation studies to show the benefits of dataset in improving existing model performance

**Questions:**

- “High-quality” dataset is over-claimed since most of the descriptions were generated by GPT4o, which has been proven to be unreliable for medical tasks. How to ensure the quality (e.g., correct disease description) of the dataset? Please also provide some sample cases for review during rebuttal.

- In figure 1. There is an obvious error in the answer to X-ray image. The box is on the left of the figure, which corresponds patients’ right lung (not left). This is also a side evidence that the data quality is not that high.

- Since one of the main contributions of this work is the dataset, necessary ablation study on the effect of dataset to model performance is missed. Can LLAVA-Med obtain better performance by training again on your dataset?

- Many medical datasets have strict licenses. Do all source datasets provide licenses for derivation or re-distribution?

---

### Official Review · Reviewer_iSaV · 2024-11-03

**Soundness:** 2
**Presentation:** 3
**Contribution:** 2
**Rating:** 5
**Confidence:** 5

**Summary:**

The paper presents the GMAI-VL model, which adopts an innovative three-stage training strategy that facilitates the integration of visual and linguistic information, thereby improving performance on multimodal medical tasks. The model's architecture is designed with the specific needs of the medical field in mind, demonstrating potential in handling complex medical data. Additionally, the paper introduces the GMAI-VL-5.5M dataset, a comprehensive and diverse multimodal medical dataset that spans various medical imaging modalities and clinical departments, making a significant contribution to advancing the field of medical AI.

**Strengths:**

1. The GMAI-VL model proposed in the paper adopts an innovative three-stage training strategy, which helps the model better integrate visual and linguistic information and improve its performance on multimodal medical tasks. The architecture design of this model considers the special needs of the medical field and demonstrates its potential in handling complex medical data.
2. The GMAI-VL-5.5M dataset proposed in the paper is a comprehensive and diverse multimodal medical dataset that covers a wide range of medical imaging modalities and clinical departments. The construction of this dataset is of great significance for promoting the development of medical AI field.

**Weaknesses:**

1. Although the paper introduces a comprehensive multimodal medical dataset, GMAI-VL-5.5M, it lacks intuitive visual aids in the methodology section, particularly in the explanation of data collection and preprocessing outlined in the third part. Including a flowchart or schematic diagram could significantly enhance understanding by clearly depicting the steps involved in these processes.

2. The GMAI-VL model architecture is well-described in the paper; however, Figures 1b and 1c could benefit from additional clarification. Specifically, a more detailed explanation of the information flow and training phase described in these figures would help readers better comprehend the model’s operational dynamics.

3. The three-stage training strategy proposed is a standout feature of the model, but the paper does not provide enough detail on its implementation. The “instruction adjustment” stage, in particular, requires more elaboration. A comprehensive explanation of this stage’s underlying principles and how it enhances the model’s performance in multimodal medical tasks would strengthen the overall understanding.

4. The introduction of new benchmark tests is somewhat superficial and lacks a thorough analysis of their design details. To highlight their importance in advancing medical AI, the paper should include an in-depth description of these benchmark tests, explaining their purpose and impact.

5. While the GMAI-VL model builds on the existing LLaVA architecture, it appears to lack significant points of innovation. Emphasizing any unique features or improvements would be beneficial, alongside experimental results that demonstrate the tangible performance enhancements these innovations bring to the model.

6. The claim of strong generalization ability is made in the paper, but there is a notable absence of experimental evidence to substantiate it. Designing and presenting experiments that evaluate the model’s generalization across various medical tasks and datasets would lend credibility to this assertion.

7. Lastly, there is minimal discussion on the model’s computational efficiency, which is crucial for practical medical applications. Metrics such as inference time and memory usage are important performance indicators. Including these metrics under different hardware configurations and comparing them with those of other models would provide a more comprehensive assessment of the model’s efficiency and practicality.

**Questions:**

1. The "instruction tuning" phase in the three-stage training strategy is not clearly detailed. Could you further explain the specific operations involved in this stage and how it significantly enhances the model’s performance on multimodal medical tasks?
2. Although the construction of the dataset is comprehensive, the paper lacks a thorough discussion of the model's computational efficiency under different hardware configurations. Could you provide more performance metrics, such as inference time and memory usage, and compare them with other models?
3. What are the key innovative aspects of the training framework, and how do these innovations differentiate it from existing approaches in the field?

---

### Note · Authors · 2024-11-13

I have read and agree with the venue's withdrawal policy on behalf of myself and my co-authors.